# Efficient Radial-Shell Model for 3D Tumor Spheroid Dynamics with Radiotherapy

**DOI:** 10.3390/cancers15235645

**Published:** 2023-11-29

**Authors:** Florian Franke, Soňa Michlíková, Sebastian Aland, Leoni A. Kunz-Schughart, Anja Voss-Böhme, Steffen Lange

**Affiliations:** 1DataMedAssist Group, HTW Dresden—University of Applied Sciences, 01069 Dresden, Germany; anja.voss-boehme@htw-dresden.de (A.V.-B.); steffen.lange@tu-dresden.de (S.L.); 2Faculty of Informatics/Mathematics, HTW Dresden—University of Applied Sciences, 01069 Dresden, Germany; sebastian.aland@math.tu-freiberg.de; 3OncoRay—National Center for Radiation Research in Oncology, Faculty of Medicine and University Hospital Carl Gustav Carus, Technische Universität Dresden, Helmholtz-Zentrum Dresden—Rossendorf, 01307 Dresden, Germany; sona.michlikova@ukdd.de (S.M.); leoni.kunz-schughart@oncoray.de (L.A.K.-S.); 4Helmholtz-Zentrum Dresden-Rossendorf, Institute of Radiooncology—OncoRay, 01328 Dresden, Germany; 5Faculty of Mathematics and Computer Science, TU Freiberg, 09599 Freiberg, Germany; 6Center for Systems Biology Dresden (CSBD), 01307 Dresden, Germany; 7National Center for Tumor Diseases (NCT), Partner Site Dresden, 69120 Heidelberg, Germany

**Keywords:** spheroids, spatio-temporal mathematical modelling, cellular automaton, radial shell model, growth curve, 3D growth, radiation therapy, simulation, systems biology, tumor relapse, minimal model

## Abstract

**Simple Summary:**

Approximately 50% of patients diagnosed with cancer receive radiotherapy at least once during their disease. Experiments with sophisticated in-cellulo assays to improve radiotherapeutic outcomes are still challenging, and some critical details of tumor cell dynamics still need to be explored. To enhance the informative value of such approaches and support future therapeutic study designs, we developed an efficient mathematical model for three-dimensional multicellular tumor spheroids, which reflect microregions within a large tumor or avascular micrometastases and which are an auspicious experimental framework to pre-assess the curative effect of radio(chemo)therapy. We validate our mathematical model using experimental tumor spheroid growth data of several cell lines with and without radiotherapy and observe equal or better performance than previous models. Moreover, our model allows for efficient parameter calibration within previously reported and/or physiologically reasonable ranges. Based on this data-driven approach, we can explain the mechanism of the characteristic dynamics at small tumor volumes.

**Abstract:**

Understanding the complex dynamics of tumor growth to develop more efficient therapeutic strategies is one of the most challenging problems in biomedicine. Three-dimensional (3D) tumor spheroids, reflecting avascular microregions within a tumor, are an advanced in vitro model system to assess the curative effect of combinatorial radio(chemo)therapy. Tumor spheroids exhibit particular crucial pathophysiological characteristics such as a radial oxygen gradient that critically affect the sensitivity of the malignant cell population to treatment. However, spheroid experiments remain laborious, and determining long-term radio(chemo)therapy outcomes is challenging. Mathematical models of spheroid dynamics have the potential to enhance the informative value of experimental data, and can support study design; however, they typically face one of two limitations: while non-spatial models are computationally cheap, they lack the spatial resolution to predict oxygen-dependent radioresponse, whereas models that describe spatial cell dynamics are computationally expensive and often heavily parameterized, impeding the required calibration to experimental data. Here, we present an effectively one-dimensional mathematical model based on the cell dynamics within and across radial spheres which fully incorporates the 3D dynamics of tumor spheroids by exploiting their approximate rotational symmetry. We demonstrate that this radial-shell (RS) model reproduces experimental spheroid growth curves of several cell lines with and without radiotherapy, showing equal or better performance than published models such as 3D agent-based models. Notably, the RS model is sufficiently efficient to enable multi-parametric optimization within previously reported and/or physiologically reasonable ranges based on experimental data. Analysis of the model reveals that the characteristic change of dynamics observed in experiments at small spheroid volume originates from the spatial scale of cell interactions. Based on the calibrated parameters, we predict the spheroid volumes at which this behavior should be observable. Finally, we demonstrate how the generic parameterization of the model allows direct parameter transfer to 3D agent-based models.

## 1. Introduction

The quantification of the biological effects of (radio)therapy is crucial for designing and evaluating treatment protocols and the response prediction for in vivo tumors. Tumor spheroids are the preferential in vitro model to study the possible means of tumor suppression [1,2,3,4,5,6,7], as they reflect many characteristic features affecting tumor growth dynamics, including three-dimensional (3D) reciprocal intercellular and cell–extracellular matrix interactions. Multicellular spheroids are 3D avascular aggregates of several thousand tumor cells that mimic tumor microareas or micrometastases. Due to their 3D structure, spheroids exhibit metabolic gradients of oxygen, nutrients, and waste products, which can strongly modulate the therapy response of cells in addition to 3D cellular interactions [3,4,5,6,7,8]. Oxygen deficiency in tumors (hypoxia) is associated with substantial radioresistance [9,10,11,12]. The lack of such resistance factors is one reason that two-dimensional in vitro approaches such as the classical clonogenic survival assays reflect the therapeutic response of cancer cells in tissue comparatively poorly [13]. Despite the complexity of the incorporated cellular processes, experiments on in vitro spheroids are less laborious and ethically questionable than in vivo animal studies [14,15,16]. Spheroid behavior and response to treatment is often evaluated in terms of the development of the spheroid volume or average diameter over time, usually denoted as the growth curve. State-of-the-art long-term spheroid-based assays assess the curative potential of new treatment strategies by classifying each spheroid within a population as either controlled or relapsed due to its individual growth recovery and kinetics, respectively. The spheroid control probabilities and selected spheroid control doses are computed as analytical endpoints by averaging the therapeutic response over ensembles of spheroids for each treatment dose [17,18,19,20], analogous to the tumor control dose employed for in vivo experiments with mice [21,22]. Growth curves and spheroid control probabilities depend on the spheroid type (cell line), the size of the spheroids at the onset of treatment, and the applied dose.

While tumor spheroids provide a physiologically more realistic in vitro framework to study tumor growth dynamics and therapeutic response, these experiments are challenging, as typically hundreds of individual spheroids have to be cultured, irradiated with different doses, and subsequently monitored for a sufficient period in order to obtain statistically reliable results for a single treatment arm. Moreover, spheroids are typically composed of several spatial domains resulting from the pathophysiological metabolic gradients, e.g., the oxygen distribution. The most prominent spatial domains developing with increasing spheroid size and volume, respectively, are a secondary necrotic core surrounded by a viable rim. Both are entangled with growth dynamics and therapy response [23,24]. However, these inner structures can only be sampled with sufficient accuracy in stained histological sections, which impedes monitoring of the dynamics in individual spheroids over time.

Mathematical modeling can provide access to these dynamics, histomorphological features, and quantitative description of treatment response, and may allow the prediction of the effect of individual therapies [25]. Therefore, such models can assist in selecting the most relevant therapeutic parameters, e.g., the range of radiation doses, to design studies and optimize experimental setups. Starting with the phenomenological description of spheroid growth curves, for example by fitting the Gompertz function [26,27,28], several mathematical models have been formulated and simulated with different approaches, complexities, and objectives. Overall, the mathematical models that describe spheroid dynamics can be categorized into nonspatial and spatial models, depending on whether or not they resolve the spatial distribution of cells. Nonspatial models concentrate on free spheroid growth, i.e., spheroid growth without treatment, starting with Greenspan [29], who explored a set of ordinary differential equations (ODE) to describe the dynamics of the radius of the spheroid and its secondary necrotic core over time. Likewise, Grimes et al. [24] used an ODE model to reproduce the free (untreated) spheroid volume growth of different cell lines mainly based on the cellular doubling time and oxygen consumption rate determined in 2D culture. Using a combination of stained histological sections of spheroids and monolayer-based Seahorse extracellular flux analyses, a relationship between the oxygen consumption rate of individual cells and the necrotic radius within the spheroids was derived [23,24]. Browning et al. [30] introduced an ODE model to simulate the radius of a spheroid over time. They further added a statistical approach to estimate the effects of expected fluctuations, finding compelling evidence that WM983b spheroids have a limiting size that is independent of the initial seeding density. Alternatively, in order to resolve the spatial structure of spheroids, Brüningk et al. [1] proposed a 3D cellular automaton and reproduced growth experiments without treatment as well as experiments with combined hyperthermia and radiation treatment for a colorectal cancer spheroid model (HCT-116). Amereh et al. [31] used a coupled system of partial differential equations (PDEs) and ODE to simulate the radius as a function of time during the early formation of tumor spheroids without treatment and derived an analytical solution for their model. Ward and King [32] and Jin et al. [33] both used models that explicitly included spatial variations in cell densities and oxygen concentrations based on PDEs to study the phenomenon of growth saturation in tumor spheroids. In both instances, their results were qualitatively compared to experimental data from real-time cell cycle imaging in spheroids [33]. Paczkowski et al. [34] employed a nonspatial logistic growth model to describe the growth of homogeneous tumor spheroids and a cellular automaton model representing a two-dimensional (2D) cross-section through a 3D tumor spheroid to demonstrate that prostate cancer spheroids comprising mixed tumor cell populations have different growth kinetics and radiosensitivities than spheroid monocultures. The 2D cross-section was used to reduce the computational cost of a full 3D cellular automaton.

Typically, models of spheroid kinetics face at least one of two challenges. On the one hand, while computationally cheap, nonspatial models lack the resolution of an oxygen gradient across the spheroid required to estimate the oxygen enhancement of radioresistance. On the other hand, while models with genuine 3D resolution can provide metabolic gradients they are computationally expensive, particularly cell-based models that scale with the number of cells. This makes calibration of parameters based on experimental data very challenging, especially as the stochastic nature of cell-based models requires an ensemble of simulation runs for each setup to obtain the average dynamics.

To overcome the challenges of both nonspatial and spatial model types, we propose a radial-shell (RS) model (Figure 1) that is spatially discrete and continuous in time. This model can be seen as a special one-dimensional Markov chain representing the cells’ dynamics in and between radial shells. For this, we assume the spheroids to be rotationally symmetric, as suggested by histological sections [18,24] and exploited by several mathematical models before [1,24,29,30,31,34]. We demonstrate that this model is (i) spatially sufficiently resolved to describe the effects of radiotherapy, and (ii) has sufficient computational performance to enable multi-parametric optimization. We show that our model can reproduce free (untreated) growth and radiotherapy response for several cell lines, including estimates of the secondary necrotic core. Its agreement with the experimental data is similar to or better than the models originally proposed for the data. Notably, this is achieved without increasing the number of fit parameters compared to the previous cell-based models. Moreover, almost all model parameters correspond to physiologically meaningful entities and are calibrated within biologically realistic ranges. We further demonstrate that virtually all model parameters can be directly transferred to a corresponding 3D cell-based model, which is potentially valuable for examining stochastic deviations in single-cell behavior as well as rotational symmetry.

## 2. Radial-Shell Model (RS Model)

We present a data-driven dynamical model for the growth kinetics of a spheroid that both resolves the radial cell composition of the spheroid and at the same time is efficient enough for manageable parameter optimization; see Figure 1 for an illustration and Appendix C for details. The spheroid is assumed to be rotationally symmetric, as suggested by experimental observations [18,24]. The resulting one-dimensional (1D) dynamics are defined by the average concentrations $c_{T}\left(r,t\right)$ of type *T* cells and the partial oxygen pressure, i.e., the oxygen pressure ρ(r,t) along the radius *r* of the spheroid discretized into radial shells of width dr.

For free (untreated) growth, two cell types *T* are considered: proliferation-competent cells T=p, and membrane-defect cells T=n. In the model context, proliferation-competent cells refers to cells with the innate capacity to proliferate with a maximal proliferation rate γ given sufficient oxygen supply and free space in their vicinity. In contrast, membrane-defect cells have undergone secondary necrosis, and as a result have permanently lost their proliferative capacity. These cells do not consume oxygen, and the spheroid volume occupied by their cell bodies and related cell debris declines at a defined rate δ. We further assume as an approximation that proliferation-competent cells consume oxygen according to a constant local oxygen consumption rate *a* as long as the oxygen pressure is above a (putatively anoxic) threshold $\rho_{\mathrm{an}}$, resulting in an oxygen gradient ρ(r,t) along the radius. At low oxygen pressure $\rho \left(r,t\right)\leq \rho_{\mathrm{an}}$ (severe hypoxia), the initially proliferation-competent cells can no longer consume oxygen to maintain their energy homeostasis and proliferation potential. Consequently, they eventually lose their survival capacity and membrane integrity with a defined rate ϵ, leading to the emergence of a core of dead cells in the center of the spheroid surrounded by a rim of proliferation-competent cells (see Figure 1 for an illustration). This structure is consistent with the histomorphological observation of a viable rim and a necrotic center in median sections of spheroids with increasing size [3,7,18,24,27].

Notably, not all proliferation-competent cells in the spheroid viable rim actively proliferate simultaneously and in the specific local micromilieu. Subpopulations undergo transient cell-cycle arrest in the 3D cellular environment of the spheroid. However, these cells may re-enter the cell cycle by stimulation via sufficient competence factors, and as such can contribute to relapse. Vice versa, the viable rim may contain membrane-intact and proliferation-incompetent cells that consume oxygen. While these permanently cell-cycle arrested cells, i.e., terminally differentiated or senescent cell populations, can alter the oxygenation within the spheroid and thereby affect the adjacent proliferation-competent cells, they do not directly contribute to the treatment response in terms of control or relapse of a spheroid, as they cannot return to an active cycling phenotype. While it is straightforward to incorporate such cells in our model, we neglect this cell type in order to obtain an effective model with a minimal number of parameters.

A radial shell i∈{0,1,2,…} encompassing all points with radius r∈[i·dr,(i+1)·dr] is centered at $r_{i}=dr\cdot \left(i+1/2\right)$ and has a volume $V_{i}=4/3\pi \left({\left[i\cdot dr\right]}^{3}-{\left[\left(i+1\right)dr\right]}^{3}\right)$. Note that while the width dr of the shells is constant, their volume increases from the center i=0 outwards and scales quadratically $V_{i}∼i^{2}dr^{3}∼r_{i}^{2}dr$ for large radii $r_{i}$. The average concentration of cell type *T* in shell *i* is $c_{T}\left(r_{i}\right)\in \left[0,1\right]$ normalized such that a total concentration $\sum_{T}c_{T}\left(r_{i}\right)=1$ represents the maximal dense packing of cells. Thus, $c_{T}\left(r_{i}\right)$ is a relative cell concentration with respect to the available space in the corresponding shell *i*, and has no unit. The cell concentration added from the proliferation-competent cells T=p in shell *i* is distributed over shells i−1, *i*, and i+1 proportional to the available free space. Thus, the shell width dr sets the spatial scale for cell transport and the distribution of daughter cells from proliferation. Notably, the width dr is not merely a discretization parameter that can be chosen to be arbitrarily small, as the effective proliferation and consequent growth dynamics depend on dr. Note that proliferation is reduced when the available free space is deficient, analogous to logistic growth. Furthermore, proliferation is assumed to be inhibited when the local oxygen pressure ρ(r,t) is below a hypoxic threshold $\rho_{h}⪆\rho_{\mathrm{an}}$. Further, cells which experience oxygen pressure below the so-called anoxic threshold $\rho_{\mathrm{an}}$ become membrane-defect cells with an ‘anoxic death rate’ ϵ. The volume occupied by membrane-defect cells is reduced at a specific rate δ. These transitions reflect diverse biological processes, including various cell death mechanisms ranging from necrosis via necroptosis, ferroptosis, and classical apoptosis, which, for brevity, are not further distinguished in the model. The quantification of these processes requires the critical radius $r_{\mathrm{an}}$ at which the oxygen pressure reaches the anoxic threshold $\rho \left(r_{\mathrm{an}}\right)=\rho_{\mathrm{an}}$. This radius is determined from the oxygen profile ρ(r), which is obtained as the steady-state solution of the rotationally symmetric diffusion equation for the given distribution of cells $c_{T}\left(r_{i}\right)$, oxygen consumption rate *a*, and oxygen diffusion constant $D_{\rho }$. As a boundary condition, we assume a constant oxygen pressure $\rho_{0}$ at the surface of the spheroid at $r_{0}$ provided by the center of the outermost shell for which the total cell concentration exceeds 0.1; see Appendix D for details. Note that additional metabolic gradients such as nutrition and waste products can be incorporated into the model; for simplicity, we use oxygen here as an effective representative gradient. Finally, to maintain a dense spherical structure, an inward transport with rate λ is implemented to account for the net effect of biomechanical mechanisms present in the spheroid, such as intercellular adhesion and mechanical pressure (see Figure 1 for an illustration).

### 2.1. Dynamics with Radiotherapy

The RS model not only resolves the growth dynamics of an untreated spheroid, it also allows the incorporation of therapeutic effects relevant to overall tumor spheroid response. In particular, in addition to the radial distribution of cells $c_{T}\left(r_{i}\right)$, the model provides the corresponding oxygen profile ρ(r), which is crucial for the radioresistance of cells. Oxygen deficiency in tissues (hypoxia) is a well-known pathological radioresistance factor; oxygen-depleted tumor areas require up to three times higher radiation doses than well-oxygenated tumor areas to achieve the same therapeutic response [10,11,12]. Various models exist to describe the effect of radiotherapy on cells [35,36]. In general, these models predict the probability $S(d_{\mathrm{eff}})$ of a cell surviving irradiation with an effective dose $d_{\mathrm{eff}}$, most prominently the linear quadratic (LQ) model $S\left(d_{\mathrm{eff}}\right)=1/\exp\left(\alpha_{\mathrm{RT}}d_{\mathrm{eff}}+\beta_{\mathrm{RT}}d_{\mathrm{eff}}\right)$ [37] with the therapeutic ratio $\alpha_{\mathrm{RT}},\beta_{\mathrm{RT}}$ of the cell line-dependent radiological constants [37,38,39], where the effective dose $d_{\mathrm{eff}}\left(d,\rho \right)\leq d$ is considered a function of the actual dose *d* and local oxygen pressure ρ, reflecting the oxygen enhancement of radioresistance. A prominent choice for the oxygen enhancement ratio (OER) $d/d_{\mathrm{eff}}$ is the Alper–Howards–Flanders oxygen equation [9]. Mitotic catastrophe has been suggested as a process for radiation-induced cell death in spheroids as a consequence of lethal irreparable DNA damage [40,41,42,43]. The incorporation of a wide variety of such mechanisms for the effect of radiation into the RS-model is straightforward. As an example, and to evaluate the performance of the model, we incorporated mitotic catastrophe as recently proposed for a 3D cellular automaton [1]. When the spheroid is irradiated with dose *d*, proliferation-competent cells $c_{p}\left(r_{i}\right)$ are instantaneously converted into damaged cells $c_{d}\left(r_{i}\right)$ with probability $1-S\left(d_{\mathrm{eff}}\left(d,\rho (r_{i})\right)\right)$. Damaged cells proliferate with probability $1-P_{\mathrm{mc}}$ analogous to proliferation-competent cells, i.e., with rate $\gamma (1-P_{\mathrm{mc}})$; however, they break up with probability $P_{\mathrm{mc}}$ during proliferation, i.e., the volume of damaged cells is reduced with a rate $\gamma P_{\mathrm{mc}}$ implemented analogously to the reduction of necrotic volume. As previously assumed [1], the probability $P_{\mathrm{mc}}$ of mitotic catastrophe takes an initial value $P_{\mathrm{mc}}^{1}<0.5$ and increases to $P_{\mathrm{mc}}^{2}>0.5$ some time after treatment.

## 3. Parameter Calibration

The parameters of the RS model can be calibrated based on experimental volume growth curves of spheroids. Further, these parameters can be categorized into four groups: known cell-line independent parameters ($D_{\rho }$, $\rho_{0}$, $\rho_{\mathrm{an}}$, $dr^{*}$), set by fixed values, known cell-line specific parameters (γ, *a*, $\alpha_{\mathrm{RT}}$, $\beta_{\mathrm{RT}}$), calibrated within their reported range (Table 1 and Table 2), unknown biological parameters (ϵ, δ, $P_{\mathrm{mc}}$), and effective parameters (λ, dr) which are fitted within a reasonable range. The reported parameter ranges are taken exclusively from monolayer experiments due to a lack of systematic parameter reporting in spheroid experiments [44]; see Table 1 for a list of references. The specific relationships between the parameters observed in the monolayer experiment and those expected in the spheroid setup are discussed below.

As a simplification for the model, the oxygen diffusion constant $D_{\rho }$, oxygen pressure $\rho_{0}$ at the surface of the spheroid, and anoxic threshold $\rho_{\mathrm{an}}$ are set to fixed values of $D_{\rho }=2\times 10^{-9}\;m^{2}s^{-1}$, $\rho_{0}=100$ mmHg, and $\rho_{\mathrm{an}}=0$ mmHg according to those used in Grimes et al. [23,24] and Brüningk et al. [1]. While the oxygen pressure $\rho_{0}$ depends in principle on the experimental setup [45,46], we found that within the experimentally observed range of $\rho_{0}\in \left[80,120\right]$ mmHg the resulting dynamics are insensitive to the exact oxygen pressure $\rho_{0}$ (Table A1 and Table A2). The proliferation rate γ and the oxygen consumption rate *a* are fitted within their cell line-specific range as implied from monolayer culture doubling times in the literature, using the latter as an upper bound of the cell cycle time (Table 1). In general, a spheroid’s maximal cell proliferation rate γ is considered to be smaller than in the corresponding monolayer culture. However, the observed values are largely dependent on the specifics of the experimental setup [44]. Note that for the FaDu cell line we assume a range of ln(2)/γ∈[20,40] h, around the reported value of 30 h. In the case of the oxygen consumption rate *a*, it has been proposed that the value *a* measured for single cells is consistent with the size of the secondary necrotic core observed in histological sections of spheroids [23,24]. As a simple reference for the spatial scale, the average single cell diameter $dr^{*}=16$ µm is fixed for all cell lines to an intermediate value of the sizes reported in the literature for the cell lines of interest [17,47,48]. The model dynamics are independent of this single cell diameter $dr^{*}$, as the oxygen consumption rate is defined per cell volume and not per cell number. Neither the anoxic death rate ϵ nor the rate δ at which the volume occupied by membrane-defect cells is reduced have been directly measured in experiments for these cell lines, and as such are fitted using a range of several orders of magnitude around the proliferation rate. The inward transport rate λ is fitted in the order of cell transport rates in tissue [49]. The shell width dr is simultaneously fitted in the range of several single cell diameters $dr=\kappa dr^{*}$. Notably, the width dr sets the spatial scale for cell transport and the distribution of daughter cells due to proliferation. In particular, the factor κ can be translated to a spatial neighborhood of a cell; see Section 4.1.

We optimized the mentioned model parameters by minimizing the residuum between the experimentally measured growth curve ${\hat{R}}_{\mathrm{spheroid}}\left(t\right)$ and the one predicted by the model $R_{\mathrm{spheroid}}\left(t\right)$ using nonlinear least-squares minimization provided by the Python package *lmfit* [50]. Because the RS model returns the radial cell concentration $c_{T}\left(r_{i}\right)$ for any time point *t* and for every cell type *T*, we define the corresponding radius of the spheroid $R_{\mathrm{spheroid}}$ via the total cell volume $\sum_{i}V_{i}\cdot \sum_{T}c_{T}\left(r_{i}\right)=4/3\pi R_{\mathrm{spheroid}}^{3}$. To take into account the inner dynamics of the spheroid, we consider the outer radius ${\hat{R}}_{\mathrm{spheroid}}$ and the secondary necrotic core simultaneously, minimizing the residuum between the radii of the secondary necrotic core based on spheroid growth experiments ${\hat{R}}_{\mathrm{necrotic}}$ and the model $R_{\mathrm{necrotic}}$. While $R_{\mathrm{necrotic}}$ is defined analogous to $R_{\mathrm{spheroid}}$ for the model, based on the volume of membrane-defect cells T=n, measuring the necrotic radius ${\hat{R}}_{\mathrm{spheroid}}$ requires histological sections, which impedes monitoring those dynamics over time in individual spheroids. A method for estimating the oxygen consumption rate *a* from the outer radius ${\hat{R}}_{\mathrm{spheroid}}$ and necrotic radius ${\hat{R}}_{\mathrm{necrotic}}$ has been devised previously (Equation (Equation 42)), and these estimates seem to be consistent with extracellular flux measurements of the oxygen consumption rate [23,24]; thus, we reverse this method to infer the corresponding necrotic radii ${\hat{R}}_{\mathrm{necrotic}}\left(t\right)$ from the growth curve ${\hat{R}}_{\mathrm{spheroid}}\left(t\right)$ using published values for the consumption rate *a*. Notably, while the radii of the spheroid $R_{\mathrm{spheroid}}$ and $R_{\mathrm{necrotic}}$ are similar in general, they are not equal to the radii $r_{0}$ and $r_{\mathrm{an}}$ from the oxygen profile. We define t=0 d as the first day of spheroid monitoring, when the spheroid has already formed. However, it needs to be emphasized that experiments usually start with the seeding of cells a few days prior. The seeding procedure and forming of the spheroid may differ between experiments; these details are neglected in our model [31]. Finally, the initial condition for the model reflecting an in vitro spheroid with ${\hat{R}}_{\mathrm{spheroid}}\left(t=0\right)$ is set by the following procedure: the radial distribution of cell concentrations $c_{p}\left(r_{i}\right)$ is initialized with a generic sigmoid function corresponding to a spheroid with much smaller radius, e.g., reflecting only 20% of the initial experimental spheroid volume, such that there is no anoxic core; subsequently, the model dynamics are integrated until the simulated spheroid reaches the size of the experimental one at t=0, and the corresponding distribution of cell concentrations is used as initial condition (see Appendix K for details).

Subsequently, the parameters $\alpha_{\mathrm{RT}}$, $\beta_{\mathrm{RT}}$, and $P_{\mathrm{mc}}$ related to radiotherapy are calibrated: the cell-specific radiosensitivities $\alpha_{\mathrm{RT}}$ and $\beta_{\mathrm{RT}}$ are taken from the ranges previously reported from clonogenic survival assays [37,38,39]. The initial and final probabilities of mitotic catastrophe $P_{\mathrm{mc}}^{1}$ and $P_{\mathrm{mc}}^{2}$ are optimized based on the dynamics of the outer radius observed at high doses, e.g., 10 Gy for HCT-116 and 20 Gy for FaDu, for which spheroids do not relapse anymore. This approach is analogous to the calibration of the other model parameters, except that only $P_{\mathrm{mc}}^{1}$ and $P_{\mathrm{mc}}^{2}$ are optimized while the other parameters are fixed to the values calibrated based on untreated growth.

For the purposes of comparison, the parameters used in the models proposed along with the experimental data are listed in Table 1 and Table 2. For the cell-based model of HCT-116, only the probabilities of necrosis ϵ·dt and the reduction of the volume occupied by membrane-defect cells δ·dt are reported [1]. For a Monte Carlo step size of dt=1 h, this translates to the corresponding rates displayed in Table 1. In the nonspatial model [24], the whole secondary necrotic core is removed within a time step dt=ln(2)/γ, from which we estimate a lower bound for the product ϵ·δ≥γ.

**Table 1 cancers-15-05645-t001:** The RS model reproduces experimental growth curves with parameters in a realistic biological range. For each experiment, the estimated parameter values and fit result of the calibrated RS Model, literature values from monolayer experiments for the parameters, and fit results from previous models (except for FaDu, for which no model has been proposed) are displayed. The parameters (left to right) are: doubling time ln(2)/γ with the proliferation rate γ, oxygen consumption rate *a*, anoxic death rate ϵ, rate δ at which the volume occupied by membrane-defect cells is reduced, the width of the radial shells $dr=\kappa dr^{*}$ expressed in multiples κ of the cell diameter $dr^{*}$, and the inward transport rate λ, displayed as the velocity λ·dr. The goodness of fit is assessed by the coefficient of determination $R^{2}$ for the volume $V_{\mathrm{spheroid}}$. Note that the two sets of lines on HCT-116 refer to separate experiments which exhibit different spheroid growth curves, probably due to discrepancies in the experimental setup. For the FaDu cell line, we assume a range of ln(2)/γ∈[20,40] h, which is around the reported value of 30 h.

Cell Line	Parameter Estimation	ln(2)/γ [h]	*a* [mmHg/s]	ϵ [1/h]	δ [1/h]	κ	λdr [µm/h]	$R^{2}$
HCT-116 Figure 2	RS-model	22.8	27.7	2.30 × 10^−1^	1.11 × 10^−2^	1.12	38.8	0.997
	Lit. (Experiment)							
	Schulte Am Esch et al. [51]; Cowley et al. [52]; Petitprez et al. [53]; Jain et al. [54]	17.1–36		-	-	-	-	-
	Grimes et al. [23]		22.1±4.8					
	Lit. (3D model)	28	22.1	∼2.78 × 10^−6^	∼2.78 × 10^−7^	∼2.04	∼12.0	0.981
	Brüningk et al. [1]							
HCT-116 Figure A1	RS-model	29.7	33.2	1.5 × 10^−1^	6.2 × 10^−3^	2.39	46.1	0.997
	Lit. (Experiment)							
	Schulte Am Esch et al. [51]; Cowley et al. [52]; Petitprez et al. [53]; Jain et al. [54]	17.1–36		-	-	-	-	-
	Grimes et al. [24]		27.9±6.0					
	Lit. (Non-spatial model)	51.8–88.6	21.87–33.97	≳γ=0.04		-	-	0.968–0.998
	Grimes et al. [24]							
MDA-MB-468 Figure A2	RS-model	52.4	15.6	4.7 × 10^−2^	3.7 × 10^−7^	9.81	4.2	0.994
	Lit. (Experiment)							
	Finlay-Schultz et al. [55]; DSMZ (www.dsmz.de)	<24, 30–80		-	-	-	-	-
	Grimes et al. [24]		18.1±4.5					
	Lit. (Non-spatial model)	45.8–62.6	13.6–22.6	≳γ=0.01		-	-	0.893–0.971
	Grimes et al. [24]							
LS-174T Figure A3	RS-model	34.7	22.5	4.5 × 10^−2^	2.0 × 10^−5^	6.22	67.8	0.998
	Lit. (Experiment)							
	DSMZ (www.dsmz.de)	30–40		-	-	-	-	-
	Grimes et al. [24]		20.6±4.4					
	Lit. (Non-spatial model)	25.68–39.12	16.2–25.0	≳γ=0.02		-	-	0.956–0.998
	Grimes et al. [24]							
SCC-25 Figure A4	RS-model	36.9	11.9	1.5 × 10^−1^	2.0 × 10^−2^	2.85	11.0	0.993
	Lit. (Experiment)							
	Steinbichler et al. [56]; Gavish et al. [57]	32.8–57.6			-	-	-	-
	Grimes et al. [24]		11.2±4.6					
	Lit. (Non-spatial model)	88.1–103.0	6.6–15.8	≳γ=0.02		-	-	0.884–0.968
	Grimes et al. [24]							
FaDu Figure A5	RS-model	36.0	9.9	3.6 × 10^−1^	3.1 × 10^−2^	4.15	53.3	0.998
	Lit. (Experiment)							
	DSMZ (www.dsmz.de)	20–40		-	-	-	-	-
	Leung et al. [58]		∼10.6					

**Table 2 cancers-15-05645-t002:** The RS model reproduces the experimentally observed spheroid dynamics after radiation for HCT-116 and FaDu. For HCT-116, the same radiosensitivities as in Brüningk et al. [1] were chosen and the probabilities of mitotic catastrophe $P_{\mathrm{mc}}$ were optimized based on the highest dose of 10 Gy while adopting the other parameters for untreated growth (Table 1). Similarly, for FaDu the probabilities of mitotic catastrophe were optimized for 20 Gy, while apt radiosensitivities were chosen from the range observed in previous clonogenic survival assays [17,59] and optimized for 5 Gy. The goodness of fit was assessed using the coefficient of determination $R^{2}$ for the volume $V_{\mathrm{spheroid}}$. The bold $R^{2}$ values indicate the experiments used for validation (not included in the calibration).

	$P_{\mathrm{mc}}^{1}$	$P_{\mathrm{mc}}^{2}$	$\alpha_{\mathrm{RT}}$	$\beta_{\mathrm{RT}}$	*d* [Gy]	$R^{2}$
HCT-116 RS-model	0.27	0.67	0.5	0.042	10 (100% control)	0.9852
					5 (100% relapse)	**0.9970**
					2 (100% relapse)	**0.9916**
HCT-116 Brüningk et al. [1]	0.44	0.59			10 (100% control)	0.98
					5 (100% relapse)	**0.88**
					2 (100% relapse)	**0.98**
FaDu RS-model	0.16	0.81	0.35	0.079	20 (100% control)	0.9867
					5 (100% relapse)	0.9812
					2.5 (100% relapse)	**0.9873**

## 4. Results

In order to demonstrate the feasibility and performance of the RS model, we first calibrated the model for five different cancer cell lines (colorectal HCT-116 and LS-174T, breast MDA-MB-468, squamous cell SCC-25, and pharynx carcinoma FaDu) from six individual experiments of untreated growth [1,18,24] and compared the goodness of fit with the models previously proposed for these experiments (Table 1). The experiments included the cell lines HCT-116, LS 174T, MDA-MB-468, and SCC-25, which have been previously compared to a nonspatial model [24], HCT-116 with a comparison to a cellular automaton model [1], and FaDu, for which no corresponding model has been proposed [18]. The goodness of fit was quantified by the coefficient of determination $R^{2}$ with respect to the spheroid volume, as reported for previous models [24]. In all cases of untreated growth, the RS model yields similar or higher goodness of fit than the corresponding models proposed for these experimental data. As an example, experimental spheroid radii ${\hat{R}}_{\mathrm{spheroid}}\left(t\right)$ and model predictions $R_{\mathrm{spheroid}}\left(t\right)$ are displayed for HCT-116 in Figure 2a, while the corresponding figures for the other cell lines can be found in Figure A1, Figure A2, Figure A3, Figure A4 and Figure A5 along with the same results reported in terms of spheroid volume. Furthermore, while the goodness of fit is only reported with regard to the spheroid radius, the RS model is additionally calibrated to reproduce the dynamics of the secondary necrotic core (Figure 2a). Note that in the RS model the secondary necrotic core is a feature emerging from the dynamics within the spheroid, while for the nonspatial model proposed for the other cell lines [24] the necrotic radius $R_{\mathrm{necrotic}}$ is simply identified with the anoxic radius $r_{\mathrm{an}}$. In particular, the distribution of membrane-defect and proliferation-competent cells over the radius of the spheroid is resolved, as shown by the examples of the first and last time point in Figure 2b,d; see Appendix A for the full dynamics over time.

Subsequently, the model was applied to experimental results after radiotherapy, available for HCT-116 at doses of 2,5,10 Gy [1] and for FaDu at doses of 2.5,5,20 Gy [18]; see Table 2 for the parameters and goodness of fit analogous to Table 1. For HCT-116, the RS model reproduces the dynamics of the spheroid after treatment as well as the model proposed along with the data. Note that only the probabilities for mitotic catastrophe $P_{\mathrm{mc}}$ are additionally calibrated based on the growth curve at 10 Gy (Figure 3), while the data for 2 and 5 Gy validate the model prediction (Figure 4 and Figure 5). The RS model reproduces the experimental results for FaDu with goodness of fits comparable to the case of HCT-116 (Figure A6, Figure A7 and Figure A8). Because a very wide range of radiosensitivities $\alpha_{\mathrm{RT}}$, $\beta_{\mathrm{RT}}$ has been reported in the literature for FaDu [17,59], we tried to choose apt values with respect to the dynamics at 5 Gy, which is the dose most sensitive to these parameters.

We highlight that the spatial resolution and agreement with the experimental results is achieved without increasing the number of fit parameters compared to the cell-based model. All the model parameters are calibrated within previously reported or physiologically reasonable ranges; see the comparison with the values reported in the literature in Table 1. Considering the challenge of parameter identifiability typical of data-driven models, the calibrated parameters reveal additional insights into cell-specific characteristic of spheroid dynamics. From the analytic solution of an ODE model of the effective spheroid dynamic, it can be inferred that the slope of a typical growth curve is determined by the product of the proliferation rate γ and shell width $dr=\kappa dr^{*}$, which sets the spatial scale for cell transport and distribution of daughter cells from proliferation (see Appendix C and Figure A11). This means that at intermediate-to-large spheroid sizes $R_{\mathrm{spheroid}}>3\cdot dr$, for which effectively only an outer shell of the spheroid is able to proliferate, the proliferation rate (γ in the case of untreated growth and $\gamma (1-2P_{\mathrm{mc}})$ after radiation) and dr cannot be inferred individually from the growth curves, only from their mutual product. However, the radial dynamics are exponential at a small spheroid radius $R_{\mathrm{spheroid}}∼dr$. The exponential phase corresponds to the scenario in which all proliferation-competent cells within the spheroid can proliferate and the exponent is essentially set by the proliferation rate. This behavior is particularly pronounced for FaDu after strong radiation for $R_{\mathrm{spheroid}}<100$ µm (Figure A6), and to some degree for MDA-MB-468 (Figure A2. This not only allows us to determine $dr=\kappa dr^{*}$ and γ independently for these cases, it demonstrates the biological relevance of the spatial scale dr. In particular, for the two cases of MDA-MB-468 and LS-174T with relatively large κ, there is a lower bound for κ≤5.4 and ≤5.7, respectively, assuming the largest proliferation rate within the range observed in monolayer, i.e., ln(2)/γ>30 h for both cell lines. Thus, we predict that for these two cell lines the exponential growth regime should be observable for spheroid sizes in the range of $R_{\mathrm{spheroid}}\in \left[dr,3dr\right]\approx \left[90,260\right]$ µm, which is currently slightly below the experimentally observed sizes. Figure A12 exemplifies the influence of different pairs of the proliferation rate γ and parameter κ for the FaDu cell line with a 20 Gy irradiation dose.

### 4.1. Parameter Transfer to a Cellular Automaton

The parameters of the RS model can be translated to a cell-based model which reflects analogous cell mechanisms, as the processes incorporated in the RS model are generic. i.e., cells consume oxygen with a rate *a* and proliferate with a rate γ given sufficient oxygen and free space in their neighborhood. The size of the directionally unbiased interaction neighborhood considered for placing potential offspring of a cell in the cellular automaton is directly related to the shell width of the RS model. In addition, cells with insufficient oxygen supply become membrane-defect cells at a rate ϵ, the volume occupied by these membrane-defect cells is reduced with a rate δ, and all cells move inwards to keep the spheroid composition dense. A cell-based model calibrated in this way displays similar dynamics on average as the RS model while additionally exhibiting statistical variation originating from the stochastic nature of the cell-based processes. This can be used to theoretically predict spheroid control probabilities and spheroid control doses for irradiation therapies, where spheroid relapse and control are distinguished and can serve as as basis to optimize experimental design for in vitro dose ranges and potentially fractionated radiation.

To demonstrate parameter transfer, we chose a cellular automaton as an example, similar to that of Brüningk et al. [1] (see Appendix I for details) and used the FaDu cell line. For brevity, we focused on untreated growth. As explained above, almost all parameters of the RS model can be directly transferred to the cellular automaton; the only exceptions are the inwards transport rate λ and shell width $dr=\kappa dr^{*}$, which are set through multiples κ of a single cell diameter $dr^{*}$. For technical reasons involving computational speed (see Appendix H for details), we chose to replace the inwards transport rate with a rate-independent inwards shuffle leading to the requirement that the Monte Carlo time step length dt of the cellular automaton be sufficiently small; see Appendix I for details.

Additionally, the shell width dr in the RS model, and consequently κ, defines the size of the interaction neighboorhood with the mean distance $d_{\mathrm{neigh}}$ in the 3D cellular automaton, i.e., it determines the maximal distance at which a cell’s daughter cell can be placed. In contrast to classical cellular automata, we allow higher orders of the typical Moore and von Neumann neighboorhoods and random alternations between them, where the order *k* is a suitably chosen integer. We demonstrate that there is an almost perfect linear dependency between κ and the mean cell distance $d_{neigh}$ of a chosen interaction neighborhood $\kappa \approx 1.19d_{neigh}-0.29$; see Figure A14 and Table A4 for more details. We use this as a basis to specify an interaction neighborhood for the example of the FaDucell line; a neighborhood which alternates 50:50 between a Moore neighboorhood of order k=3 ($d_{\mathrm{neigh}}=3.34$) and k=4 ($d_{\mathrm{neigh}}=4.31$) corresponds to κ≈4.26, which is close to the value of κ=4.31 calibrated with the RS model for FaDu. We find good agreement for the radii $R_{\mathrm{spheroid}}$ and $R_{\mathrm{necrotic}}$ over time between both models as well as for the cell concentrations (Figure 6).

## 5. Discussion

We propose an effectively 1D radial-shell (RS) model for simulating 3D tumor spheroid dynamics with and without radiotherapy while taking into account the oxygen-dependent radiosensitivity of cells. By deliberately exploiting the approximate radial symmetry of spheroids as evident in the data, we are able to considerably reduce the computational complexity of our model without compromising the quality of agreement between model and data compared to previous genuinely 3D cell-based models. We show that the proposed RS model reproduces untreated growth and radiotherapy response experiments for many different cell types and doses, including estimates of the secondary necrotic core, while restricting the parameters to biologically realistic ranges. At the same time, our model has sufficient computational performance to allow for multi-parametric optimization, enabling fits which are similar or better than those obtained from models originally proposed for the data. Additionally, we have shown through examples that there is a parameter mapping between the new RS model and related 3D cell-based models that leads to full agreement of the two models with respect to common experimentally relevant observables. In this way, we open up a way for efficient multi-parametric optimization of those cell-based models while at the same time demonstrating the adequacy of dimensional reduction.

The RS model is an effective implementation of key cell-based mechanisms underlying tumor spheroid dynamics and oxygen-dependent therapy response that is parameterized according to biological interpretables and to some extent experimentally identifiable quantities. In particular, we show that the shell size $dr=\kappa dr^{*}$ in the RS model translates to the size of the neighborhood in a 3D cellular automaton, and as such can be biologically interpreted as an effective range of proliferation. To guide experimental identification, we analytically predict the influence of the effective range dr on different spheroid growth regimes in the model; exponential growth for small radii is superseded by linear growth for larger radii, and the position of the area of transition is essentially determined by the effective range of proliferation (or equivalently, the shell size). Further, the increase in the exponential phase is set by the proliferation rate alone, while the gradient of the linear growth is determined by the product of proliferation rate and effective range of proliferation, which means that parameter identification is enhanced if observation of at least a few small radii can be included in the calibration process. The effect of the range of proliferation dr is observable in the experimental data for FaDu spheroids after strong irradiation. While the spheroids in the available experiments for the other considered cell lines are too large to exhibit the effect of dr, we predict that this effect should be experimentally observable for MDA-MB-468 and LS-174T. Based on our calibrations for these two cell lines, we estimate a range of spheroid diameters 180–520 µm in which the transition between linear and exponential growth regime is expected, which could be validated by future experiments.

It is known that there are several other cell states and intercellular processes present in a spheroid beyond those incorporated into the presented RS model, including proliferation-incompetent yet oxygen consuming membrane-intact cells, additional metabolic gradients including waste products, cell cycle arrest, and complex pathways for cell death and destruction due to anoxia or radiation. However, our goal is not to reflect all known biological processes within a spheroid but to propose an effective model which, as demonstrated, is sufficient to reproduce the available experimental growth curves based on a few key mechanisms. The model can be easily extended with additional cellular mechanisms if systematic data related to these mechanisms are available. Moreover, it can be used as a framework to include and theoretically analyze the potential effects of additional determinants, such as genetic heterogeneity of the cells modifying their motility, growth, and radioresponse characteristics, cellular plasticity [60,61], stochasticity in cellular response behavior, and loco-regional and spatio-temporal pathophysiological phenomena of tumor tissues such as chronic and acute hypoxia. While it is straightforward to incorporate additional biological processes into the RS model (e.g., the constant outer oxygen pressure $\rho_{0}$ may be replaced by a time-dependent function $\rho_{0}\left(t\right)$ to mimic acute and intermittent hypoxia), these extensions require the introduction of additional parameters. This is only sensible if these parameters are already known from independent experiments or if their impact is observable in complementary measurements which can be compared to simulation results, e.g., stained histological sections [23,24,46,62], cell counts [46,62,63], cell cycle imaging [33], or genetic heterogeneity. Furthermore, while estimates of several cell line-specific parameters can be found in the literature, e.g., for proliferation rates ([51,52,53,54,55,56,57] and DSMZ (www.dsmz.de)), most of these originate from monolayer experiments. Moreover, the effective values of such parameters in spheroid experiments depend not only on the cell line but on the details of the experimental setup, e.g., oxygenation, nutrition concentration, and seeding procedure. This becomes evident when comparing the two growth experiments without treatment on HCT-116 [1,24], which display considerably different spheroid growth curves and consequently result in different estimated proliferation rates for the RS model. This highlights the importance of systematic high quality spheroid experiments with detailed parameter reporting [44].

In the future, we plan to apply and verify the RS model in several scenarios. First, the parameters can be calibrated based on spheroids of a particular initial size, with the calibrated model consequently used to predict the untreated growth and therapy response of larger or smaller spheroids. This can extend the informative value of spheroid experiments and represent a first step in statistical projection from spheroid control probabilities to tumor control probabilities, as spheroids may be considered avascular microareas of varying size within a macroscopic tumor. Second, the parameters resulting from calibration of the RS model can help to uncover potential differences between spheroid types and experimental conditions as a way to stimulate extended studies and guide future experimental design. For instance, the anoxic death rate, the rate at which the volume occupied by membrane-defect cells is reduced, and the inwards velocity are all parameters of the RS model that can be readily interpreted in biological terms but which have not yet been directly identified experimentally. Third, the parameters can be calibrated based on spheroids treated with a single dose, and the calibrated model can then be used to predict the response to fractionated treatment. While fractionated treatment is more realistic than single-dose treatment, the corresponding in vitro experiments with spheroids are more laborious. In this context, model simulations can help to identify the relevant dose range, thereby reducing the required number of experiments. Finally, our results suggest that the proposed model has the potential to guide the design of future experiments at experimentally unseen doses. We have demonstrated for two examples that, following calibration based on curves from untreated growth and at high radiation dose, the model is able to predict the therapy response at intermediate doses for given radiosensitivities $\alpha_{\mathrm{RT}}$ and $\beta_{\mathrm{RT}}$ from monolayer experiments. A prerequisite for this application is the determination of effective radiosensitivities in spheroids and their relation to the corresponding monolayer values. While it should be possible to fit such effective radiosensitivities based on at least two growth curves at intermediate doses, a more promising route would be to independently extract these values from spheroid control probabilities using a suitable statistical model.

The proposed RS model framework stands out through its computational efficacy while maintaining high conformance to genuine 3D cell-based models. It can serve as an expedient means to theoretically analyze and question experimental data by providing estimates of important influencing factors and allowing extrapolation to experimentally unexamined conditions. In particular, it could be used in the future to optimize fractionation and scheduling of single or multi-modality treatments as well as to explore the consequences of competing hypotheses on the cellular therapy response in order to gain understanding and guide future experiments.

## Figures and Tables

**Figure 1 cancers-15-05645-f001:**
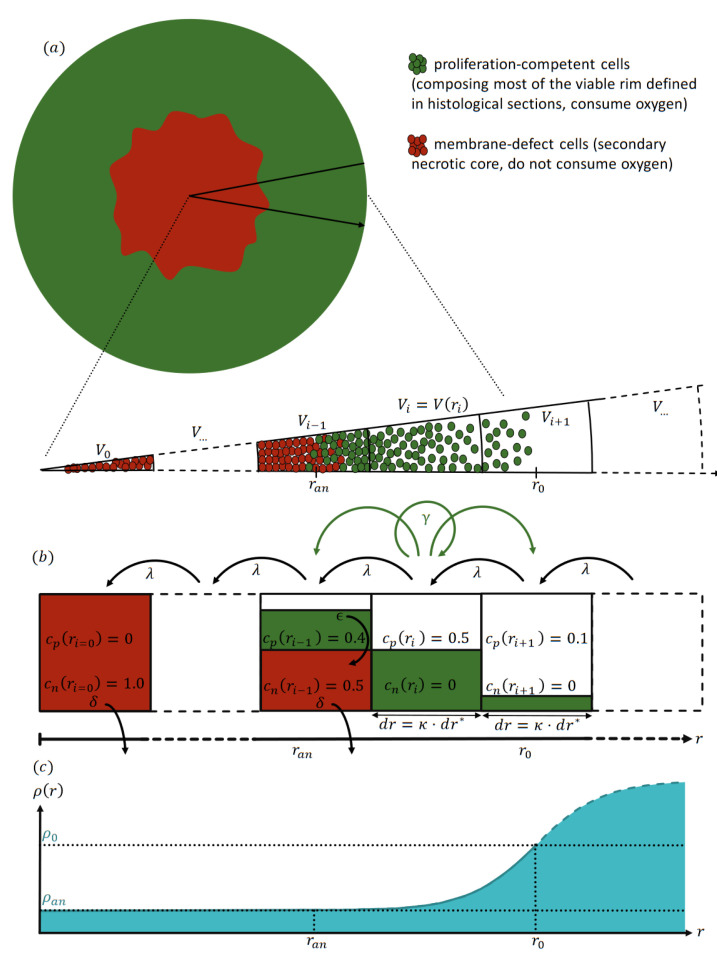
Illustration of the radial-shell model (RS-model). (**a**) Sketch of a typical stained histological section of a spheroid with a viable rim (green) surrounding a secondary necrotic core, with cell debris membrane-defect cells (red) suggesting an approximate rotational symmetry of the spheroid. (**b**) By exploiting the symmetry, the model is able to consider the cell dynamics on radial shells i∈[0,1,2,...) and along the distance *r* from the center of the spheroid at i=0. The shells have equal radial width dr, and their maximal volume $V_{i}$ is consequently increasing outwards. Each shell *i* holds a concentration of cells $c_{T}\left(r_{i}\right)$ of type *T* cells, where T=p denotes proliferation-competent cells (green, proliferate with maximal rate γ, consume oxygen with rate *a*) and T=n membrane-defect cells (red, irreversibly lost competence to proliferate, do not consume oxygen). Note that the concentrations $c_{T}\in \left[0,1\right]$ are normalized according to the available space in each shell. (**c**) Oxygen consumption of proliferation-competent cells leads to a radial decrease of oxygen pressure from the value $\rho_{0}$ at the most outer shell at $r_{0}$ down to an anoxic threshold $\rho_{\mathrm{an}}$ at $r_{\mathrm{an}}$ towards the center of the spheroid. Beyond the threshold $\rho_{\mathrm{an}}$, cells are anoxic and become membrane-defect with the anoxic death rate ϵ. The volume occupied by membrane-defect cells is reduced with rate δ and all cells are transported inward with rate λ such that the spheroid remains compact. Then, the outer radius $R_{\mathrm{spheroid}}$ and necrotic radius $R_{\mathrm{necrotic}}$ are defined as the radius of a sphere with a volume equal to the total cell volume and membrane-defect cell volume, respectively. For reference, the shell width $dr=\kappa dr^{*}$ is expressed as multiple κ of a single cell diameter $dr^{*}$.

**Figure 2 cancers-15-05645-f002:**
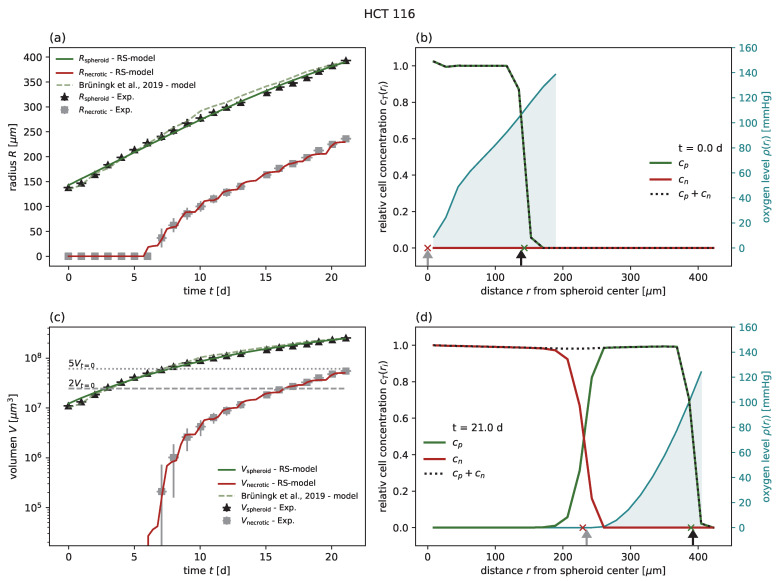
(**a**) The RS model reproduces the outer radius (dark green) for untreated growth of the cell line HCT-116, with experimental data from Brüningk et al. [1] (black crosses) even better than the model proposed along with the data (light green dashed). Note that the necrotic radius during the experiment can be estimated (gray crosses; see Section 3 for details) and that the RS model additionally reproduces this necrotic radius (red). (**c**) Volumes corresponding to the radii displayed in (**a**) in a semi-log plot. Multiples of 2 and 5 of the starting volume $V_{t=0}$ are highlighted with horizontal gray dashed and gray dotted lines, respectively. The RS model predicts the cell concentration over the radius at any time point. Examples show the cell distributions for (**b**) the first time point t=0 d and (**d**) the last day t=21 d of the experiment. Displayed are the cell concentrations of proliferation-competent cells $c_{p}$ (green solid), membrane-defect cells $c_{n}$ (red solid), total concentration of cells (black dotted), and oxygen profile (blue solid, right y axis). The marker at the bottom of the panels (**b**,**c**) indicates the necrotic and outer radius estimated from the model (red and green crosses) and from the actual experimental data (gray and black arrows), respectively. The used parameters are reported in Table 1.

**Figure 3 cancers-15-05645-f003:**
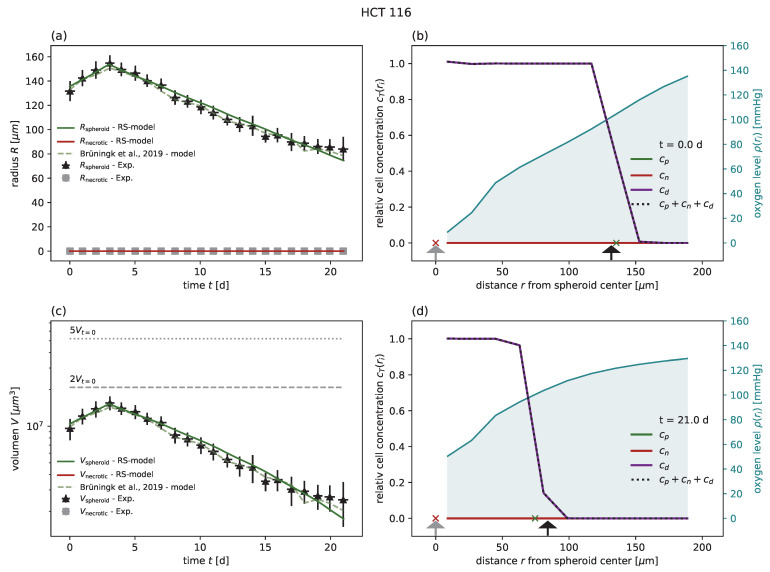
(**a**) The RS model reproduces the outer radius (dark green) for the cell line HCT-116 with 10 Gy radiation experimental data from Brüningk et al. [1] (black crosses) as effectively as the model proposed along with the data (light green dashed). We used the parameters calibrated for untreated growth, only fitted the probabilities of mitotic catastrophe $P_{\mathrm{mc}}^{1}$, $P_{\mathrm{mc}}^{2}$, and adopted the radiosensitivities $\alpha_{\mathrm{RT}}$ and $\beta_{\mathrm{RT}}$ from Brüningk et al. [1]. Note that for this regime no secondary necrotic core is expected (experimental estimate shown as gray crosses), which is reproduced by the RS model (red). The parameters we used can be found in Table 1 and Table 2. (**c**) Volumes corresponding to the radii displayed in (**a**) in a semi-log plot. Multiples of 2 and 5 of the starting volume $V_{t=0}$ are highlighted with horizontal gray dashed and gray dotted lines, respectively. The RS model predicts the cell concentration over the radius at any time point. Examples show the cell distribution for (**b**) the first time point t=0 d and (**d**) the last day t=21 d of the experiment. Displayed are the cell concentrations of proliferation-competent cells $c_{p}$ (green solid) and membrane-defect cells $c_{n}$ (red solid), the total concentration of cells (black dotted), and the oxygen profile (blue solid, right y axis). The marker at the bottom of the panels (**b**,**c**) indicates the necrotic and outer radius estimated from the model (red and green crosses) and from the actual experimental data (gray and black arrows), respectively.

**Figure 4 cancers-15-05645-f004:**
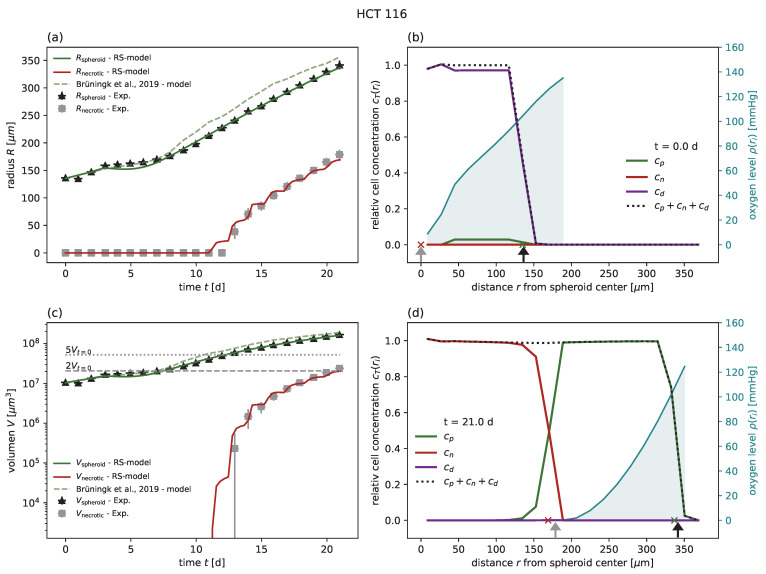
(**a**) The RS model reproduces the outer radius (dark green) for the cell line HCT-116, with 5 Gy radiation experimental data from Brüningk et al. [1] (black crosses) even better than the model proposed along with the data (light green dashed). We used the parameters calibrated for untreated growth, the values for $P_{\mathrm{mc}}^{1}$ and $P_{\mathrm{mc}}^{2}$ fitted at 10 Gy, and the values $\alpha_{\mathrm{RT}}$ and $\beta_{\mathrm{RT}}$ from Brüningk et al. [1]. Note that the necrotic radius during the experiment can be estimated (gray crosses) (see Section 3 for details) and that the RS model reproduces this necrotic radius (red). Additionally, note that the necrotic radius of the RS model appears jagged due to the spatial discretization of $r_{0}$. The used parameters are reported in Table 1 and Table 2. (**c**) Volumes corresponding to the radii displayed in (**a**) in a semi-log plot. Multiples of 2 and 5 of the starting volume $V_{t=0}$ are highlighted with horizontal gray dashed and gray dotted lines, respectively. The RS model predicts the cell concentration over the radius at any time point. Examples show the cell distribution for (**b**) the first time point t=0 d and (**d**) the last day t=21 d of the experiment. Displayed are the cell concentrations of proliferation-competent cells $c_{p}$ (green solid) and membrane-defect cells $c_{n}$ (red solid), the total concentration of cells (black dotted), and the oxygen profile (blue solid, right y axis). The marker at the bottom of the panels (**b**,**c**) indicates the necrotic and outer radius estimated from the model (red and green crosses) and from the actual experimental data (gray and black arrows), respectively.

**Figure 5 cancers-15-05645-f005:**
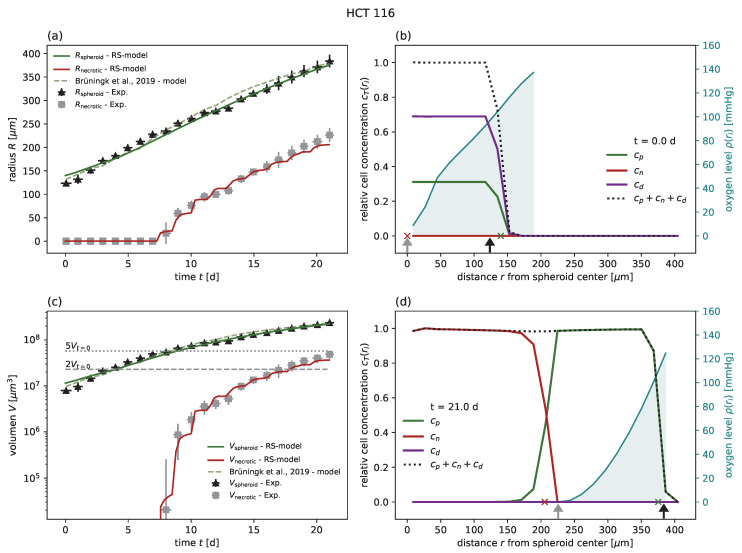
(**a**) The RS model reproduces the outer radius (dark green) for the cell line HCT-116, with 2 Gy radiation experimental data from Brüningk et al. [1] (black crosses) as effectively as the model proposed along with the data (light green dashed). We used the parameters calibrated for untreated growth, the values for $P_{\mathrm{mc}}^{1}$ and $P_{\mathrm{mc}}^{2}$, and the model proposed along with the data (light green dashed) at 10 Gy, as well as the values $\alpha_{\mathrm{RT}}$ and $\beta_{\mathrm{RT}}$ from Brüningk et al. [1]. The necrotic radius during the experiment can be estimated (gray crosses) (see Section 3 for details) and the RS model reproduces this necrotic radius (red). Note that the necrotic radius of the RS model appears jagged due to the spatial discretization of $r_{0}$. The used parameters can be found in Table 1 and Table 2. (**c**) Volumes corresponding to the radii displayed in (**a**) in a semi-log plot. Multiples of 2 and 5 of the starting volume $V_{t=0}$ are highlighted with horizontal gray dashed and gray dotted lines, respectively. Additionally, the RS model predicts the cell concentration over the radius at any time point. Examples show the cell distribution for (**b**) the first time point t=0 d and (**d**) the last day t=21 d of the experiment. Displayed are the cell concentrations of proliferation-competent cells $c_{p}$ (green solid) and membrane-defect cells $c_{n}$ (red solid), the total concentration of cells (black dotted), and the oxygen profile (blue solid, right y axis). The marker at the bottom of the panels (**b**,**c**) indicates the necrotic and outer radius estimated from the model (red and green crosses) and from the actual experimental data (gray and black arrows), respectively. The used parameters are reported in Table 1.

**Figure 6 cancers-15-05645-f006:**
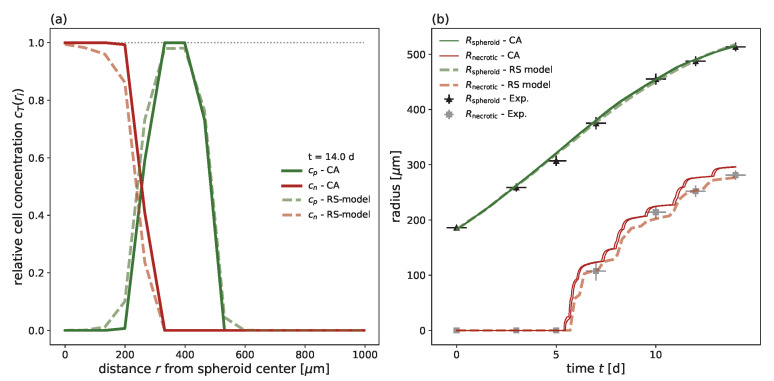
The cellular automaton reproduces the experimental growth curve for the FaDu cell line after parameter transfer from the previously fitted RS model. We directly transferred the parameters of the RS model to the cellular automaton using a 50:50 alteration between the 3-Moore and 4-Moore neighborhoods to achieve a value of κ=4.26. (**a**) Cell concentration over the radius at t=14 d for the 3D cellular automaton; the mean cell concentrations (solid) from ten independent simulation runs are compared with the cell concentrations of the corresponding RS model (dashed). The cell concentrations of proliferation-competent cells $c_{p}$ (green solid) and membrane-defect cells $c_{n}$ (red solid) are displayed. (**b**) Growth dynamics over time for the outer radius $R_{\mathrm{spheroid}}$ (green) and necrotic radius $R_{\mathrm{necrotic}}$ (red) along with the experimental outer radius (black crosses) and corresponding estimated necrotic radius (gray crosses). For the cellular automaton, the maximal and minimal radii from ten independent simulations are shown.

## Data Availability

The code is publicly available under https://gitlab.florian-franke.eu/florian/radialShell_and_cellular_automaton and https://github.com/0815IDIOT/radialShell_and_cellular_automaton (accessed on 21 November 2023).

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
