# Peer review of "Efficient Radial-Shell Model for 3D Tumor Spheroid Dynamics with Radiotherapy"

_cancers, 2023, doi:10.3390/cancers15235645_

Round 1
Reviewer 1 Report
Comments and Suggestions for Authors
In the work entitled: "Efficient radial-shell model for 3D tumor spheroid dynamics with radiotherapy“, the authors report on a radial-shell model of 3D cancer spheroids that is spatially discrete and temporally continuous. The work is complex and very well elaborated. The authors have supported their calculations with experimental work. The model takes into account the oxygen supply to the tumour, which changes towards the core of the spheroid, which is often hypoxic. The development of spheroid growth as a function of cell type and irradiation dose is demonstrated.
The work presented in the manuscript is valuable and brings new insights to predict and understand the response of cancer cells in 3D models to the treatment modality.
There are only a few minor comments:
The text in Table 1 is small and not easy to read.
There are typos in the references on page 16, lines: 460, 465.
Reviewer 2 Report
Comments and Suggestions for Authors
Comments to authors:
In this manuscript, authors have developed a one-dimensional mathematical model that shows the equal or better performance compared to the published 3D dynamics agent-based models. This redial-shell model can incorporate the 3D dynamics of tumor spheroids and reproduce the experimental spheroid growth curves. Compared to the typical 3D tumor spheroid models, RS model shows the computationally cheap and non-heavily parameterized, performs the multi-parametric optimization with published results or physiologically reasonable ranges. This manuscript is interesting, and meaningful. However, I just have some questions about this manuscript.
--In this model, the authors have only shown the ideal situation, however, in the clinical, tumor is pretty complexity. For example, autophagy mainly involved in promoting cancer cell motility, therefore, the number of the membrane-defect cells will be modified. So, the formulas should be calibrated by some parameters.
--In the tumor research, the research object should be the whole human body, but sometimes, a cancer patients can not only be diagnosed with cancer, but also with some another disease, such as COPD, in another word, the blood oxygen content is lower, thence the cancer cells exhibit metabolic oxygen is not enough, should the model be calibrated in this situation?
Reviewer 3 Report
Comments and Suggestions for Authors
Franke et al.'s “Efficient radial-shell model for 3D tumor spheroid dynamics with radiotherapy” manuscript describes a comprehensive radial shell model for 3D tumors. The authors have written the manuscript well and appropriately provided an in-depth literature review. The radial shell model described by the authors also recapitulates the growth of several cell lines they have checked, which is significant validation for their radial shell model. The 1D radial-shell model for stimulating 3D spheroid dynamics is an innovative approach for predicting radiotherapy responses. The authors have shown comparable computational advantages of the RS model over 3D base models used to predict experimental outcomes for spheroids. This study presents an important baseline for experimental exploitation of the non-examined condition which is a significant contribution to the therapeutics experiments based on the organoid model system. The authors have provided enough supporting data that reflects how their mathematical model depicts the growth curves of the cell lines spheroids' growth used in the study. I understand that this study is considerably complete to be published as it is in the Cancers journal.
